# Biologically-informed regional subset analysis with CatBoost for robust tissue-of-origin prediction

Sungmin Yang[1], Hong-Gee Kim [1,2]*

1 Biomedical Knowledge Engineering Laboratory, Seoul National University School of Dentistry, Seoul, Republic of Korea, 2 Dental Research Institute, Seoul National University, Seoul, Republic of Korea

* hgkim@snu.ac.kr

## Abstract

Accurate identification of cancer tissue/cell of origin (TOO/COO) is critical for diagnosis and treatment; yet existing whole-genome approaches demand extensive computation and often struggle with sparse mutation signals. Here, we introduce an informative regional subset framework that selects a small number of biologically and statistically significant 1Mbp genomic intervals to train a CatBoost prediction model. On a benchmark of 137 whole-genome samples across six cancer types, our method achieved a 4% gain in melanoma accuracy (from 88.0% using all 2,128 regions to 92.0% with 300 regions), a 4.4% gain in multiple myeloma (87.0% with 600 regions), and perfect (100%) accuracy in high-mutation cancers such as esophageal adenocarcinoma and glioblastoma with as few as 50 informative regions. When extended to 934 PCAWG samples spanning 14 cancer lineages, the same limited regional subsets matched or improved whole-genome performance, reaching up to 100% accuracy in gastrointestinal, skin, and brain cancers, demonstrating exceptional scalability. Our approach not only reduces computational burden and enhances interpretability but also provides a robust, generalizable tool for precision oncology and the diagnosis of cancers of unknown primary.

## 1 Introduction

The remarkable advances in DNA sequencing technologies have enabled the large-scale accumulation of genome-wide somatic mutation data across diverse cancer types, precancerous lesions, normal tissues, and stem cells [1]. This wealth of genomic data has progressively unveiled the complexity of cancer genomes, encompassing driver and passenger mutations, mutational signatures [2], clonal and subclonal evolution, and structural variations including kataegis, chromothripsis, and whole genome doubling.

Tissue-of-origin (TOO) and cell-of-origin (COO) prediction addresses the question: "From which normal cell or tissue did this cancer initially arise?" with clinical implications for metastatic cancers and cancers of unknown primary [3,4]. Cancers

**Data availability statement:** Section [Method] 2.1 Somatic Mutation Data Collection of the manuscript describes the data sources and provides information about the data owners. The PCAWG data can also be easily downloaded from publicly accessible services such as https://www.cbioportal.org/.

**Funding:** This work was supported by the government of the Republic of Korea (MSIT) and the National Research Foundation of Korea (NRF-RS-2023-00268071 to Hong-Gee Kim).

**Competing interests:** The authors have declared that no competing interests exist.

of unknown primary (CUP) account for approximately 3% of all cancer diagnoses, and patients suffering from this condition face significant therapeutic challenges, as primary cancer type classification is the dominant factor guiding treatment decisions. Early computational methods applied Random Forests for classification [5], later adopting XGBoost for scalable gradient boosting [6], and most recently CatBoost to leverage ordered boosting and categorical feature support [7,8]. Among these approaches, CUPLR [9] has established itself as the current state-of-the-art WGS-based classifier, achieving 90% recall across 35 cancer types by integrating complex features, including structural variant signatures, viral DNA integrations, and gene fusions. Complementary modalities have also demonstrated promise: deep neural networks trained on pan-cancer gene expression data achieve 97% accuracy with robust performance on metastatic samples [10,11], though they remain constrained by tissue availability and RNA stability limitations. DNA methylation-based machine learning models classify primary organs with 87–97% accuracy from tissue or cell-free DNA [12,13], and serum miRNA classifiers achieve up to 95% accuracy in top-3 predictions [14,15]. Histopathology-based deep learning approaches can predict tumor origin with an AUC of 0.95–0.99, occasionally outperforming experienced pathologists [16,17]. Despite these advances, these approaches face significant limitations: they often lack biological grounding, require data modalities that may not be available in all clinical contexts, and impose substantial computational burdens on clinical laboratories.

Existing genome-wide approaches, particularly whole-genome based methods, present fundamental limitations that constrain their clinical adoption and biological interpretability. Existing regional approaches such as COOBoostR [18] partition the genome arbitrarily without integrating biological context or employing systematic statistical selection criteria. These genome-wide strategies generate high-dimensional feature spaces that are computationally expensive, difficult to interpret biologically, and create practical barriers to clinical implementation. Furthermore, genome-wide models fail to leverage known cancer biology regarding tissue-specific mutational processes and chromatin organization [19–21], resulting in feature sets that may contain substantial noise and redundancy. This disconnect between computational methodology and underlying cancer biology limits confidence in model predictions and complicates clinical validation in molecular tumor board settings. Additionally, current whole-genome approaches emphasize high-dimensional feature selection without systematically identifying which genomic intervals are truly informative for cancer classification nor do they provide cancer-specific biological insights that could advance understanding of tissue-specific mutation patterns.

Here, we introduce a fundamentally different approach that addresses these critical limitations. Rather than leveraging whole-genome features, we present a novel biologically-informed regional subset framework that systematically identifies small sets of informative genomic intervals—optimized at various scales (50, 100, 200, 300, 400, 500, 600 Mbp)—guided by mutation density profiles reflecting tissue-type-specific chromatin states and chromatin accessibility features. This represents a paradigm shift from traditional whole-genome approaches: instead of using all genomic information indiscriminately, we employ rigorous biological and statistical

criteria to identify the most discriminative regional subsets that capture tissue-specific mutational signatures. Using these carefully selected regional subsets, we train an optimized CatBoost model that achieves superior performance compared to existing whole-genome methods. The key innovations of this work are threefold: (1) we demonstrate that informative regional subsets (typically representing only 2–30% of the genome) can replace genome-wide features while maintaining or improving classification accuracy, fundamentally challenging the assumption that more data necessarily yields better predictions; (2) we develop a machine learning algorithm with dramatically improved efficiency and accuracy compared to existing approaches, significantly reducing computational burden while enhancing prediction speed and reliability; and (3) we identify cancer-specific biological regional subsets that provide actionable insights into the genomic basis of tissue-of-origin classification, enabling clinicians to understand which genomic features drive predictions for specific cancer types. These regional subsets represent a novel biological resource that can inform future studies of cancer genomics and tissue-specific mutational processes. By combining biological sophistication with computational efficiency and clinical practicality, our approach represents a significant advance toward integrating precision genomics into routine cancer diagnostics.

## 2 Materials and methods

### 2.1 Somatic mutation data collection and preprocessing

**2.1.1 Benchmark dataset.** To directly compare the predictive performance of CatBoost with existing Random Forest and XGBoost–based methods, we assembled a benchmark cohort of 137 whole-genome sequencing (WGS) tumor samples across six cancer types: melanoma (ME, $n = 25$) [22], multiple myeloma (MM, $n = 23$) [23], liver cancer (RK, $n = 64$) [24], colorectal cancer (CRC, $n = 9$) [25], glioblastoma (GBM, $n = 7$) [26], and esophageal adenocarcinoma (ESO, $n = 9$) [27]. All variant coordinates were converted from GRCh38 to GRCh37 (hg19) using CrossMap [28]. Hypermutant samples and low-quality or duplicated variant calls were removed to ensure high-confidence somatic mutation profiles. This dataset served both to benchmark CatBoost performance and to inform our Informative Region Selection framework.

**2.1.2 PCAWG dataset.** For generalizability and scalability assessment, we obtained 934 tumor samples from the ICGC and PCAWG Consortium [29]. Samples were grouped into lineage cohorts analogous to the benchmark set (e.g., SKCM/MELA for melanoma; ESAD/READ/STAD for esophageal; LIHC/LIRI/LICA for liver; COAD for colorectal; CLLE/LAML for hematologic; GBM for brain; MALY for myeloma). All samples underwent identical GRCh37 liftover and filtering of low-confidence and duplicate variants.

**2.1.3 Mutation density profiling.** Autosomal chromosomes (1–22) were partitioned into consecutive 1 Mbp windows, excluding centromeric/telomeric regions, low-mappability segments (score < 0.5), and extreme GC content (< 20% or > 80%). Somatic point mutations were aggregated per window using BEDOPS [30] to generate mutation density profiles. These densities served as input features for CatBoost and as the basis for regional subset identification.

### 2.2 Chromatin mark data collection and preprocessing

We collected 673 ChIP-seq datasets of histone modifications from ENCODE [31], the International Human Epigenome Consortium [32], and the NIH Roadmap Epigenomics Project [33]. Data were binned into the same 1 Mbp windows (hg19), normalized, and aggregated to yield chromatin state feature matrices for each sample.

### 2.3 CatBoost–based tissue/cell of origin prediction

**2.3.1 Gradient boosting fundamentals.** CatBoost is a variant of gradient boosting that sequentially learns decision trees to correct prediction errors from previous trees. The objective of basic gradient boosting is to minimize the following loss function:

$$L(y, \hat{y}) = \sum_{i=1}^{n} l(y_i, \hat{y}_i) \tag{1}$$

where $y_i$ represents the true class (tissue/cell of origin), $\hat{y}_i$ is the model's prediction, and $l$ is the loss function. For our multi-class classification problem, we use multinomial cross-entropy loss:

$$l(y_i, p_i) = -\sum_{k=1}^{K} y_{i,k} \log(p_{i,k}) \tag{2}$$

where $K$ is the total number of tissue/cell origin classes and $p_{i,k}$ is the predicted probability that sample $i$ belongs to class $k$.

**2.3.2 Algorithm rationale and architecture.** A key feature of CatBoost is its effective handling of categorical variables. To encode categorical variables present in genomic metadata, CatBoost employs ordered target encoding:

$$x_{i,c}^{encoded} = \frac{\sum_{j=1}^{i-1}[x_{j,c} = x_{i,c}] \cdot y_j + a \cdot p}{\sum_{j=1}^{i-1}[x_{j,c} = x_{i,c}] + a} \tag{3}$$

where $a$ is a regularization parameter and $p$ is the prior class probability. This approach prevents target leakage by using only data prior to each sample.

**Symmetric trees.** CatBoost uses symmetric tree structures to reduce overfitting. At each split, the optimal feature $f^*$ and threshold $t^*$ are selected as follows:

$$f^*, t^* = \arg\min_{f,t} \left[ L_{\text{left}}(f, t) + L_{\text{right}}(f, t) \right] \tag{4}$$

where $L_{\text{left}}$ and $L_{\text{right}}$ are the losses of the left and right child nodes, respectively.

**Boosting procedure.** The model is trained iteratively as follows:

$$\hat{y}_m(x) = \hat{y}_{m-1}(x) + \eta \cdot h_m(x) \tag{5}$$

where $\hat{y}_m$ is the prediction after the $m$-th iteration, $h_m$ is the $m$-th tree, and $\eta$ is the learning rate. Each tree $h_m$ learns to fit the negative gradient of the previous prediction:

$$g_{i,m} = -\frac{\partial l(y_i, \hat{y}_{m-1}(x_i))}{\partial \hat{y}_{m-1}(x_i)} \tag{6}$$

We selected CatBoost [34] for its ordered boosting and permutation-driven training, which mitigates target leakage and captures sequential correlations among adjacent genomic windows. Input features comprised mutation density and chromatin signals across all 2,128 windows. A backward-elimination procedure, coupled with ten-fold cross-validation, was employed over 20 iterative training rounds to identify the top 20 predictive chromatin features.

**2.3.3 Hyperparameter optimization.** We defined the search space for hyperparameter tuning as follows:

- *Learning rate ($\eta$)*: 0.005 to 0.5, representing the contribution of each tree to the final prediction
- *Tree depth ($d$)*: 3 to 7, controlling the complexity of individual decision trees
- *Number of iterations ($M$)*: 10 to 100, determining the total number of boosting rounds

The learning rate controls the step size in the boosting update (Eq 5), with smaller values typically requiring more iterations to converge but potentially avoiding overfitting. Tree depth determines the interaction complexity among features,

with deeper trees capturing more complex patterns but increasing overfitting risk. The number of iterations determines the total boosting rounds performed in the ensemble.

We employed an exhaustive grid search combined with early stopping to identify the optimal hyperparameter configuration. The grid search evaluated all combinations of hyperparameters within the defined search space. To prevent unnecessary computation and overfitting, we implemented early stopping with a patience of 10 rounds, which terminates training if the validation loss does not improve for 10 consecutive iterations.

The optimization criterion was root mean square error (RMSE), defined as:

$$\text{RMSE} = \sqrt{\frac{1}{n}\sum_{i=1}^{n}(y_i - \hat{y}_i)^2} \tag{7}$$

where $y_i$ is the true class prediction score and $\hat{y}_i$ is the predicted score. We utilized stratified k-fold cross-validation (k=5) to ensure robust evaluation across the entire dataset and mitigate the impact of data splits.

Grid search identified the optimal hyperparameter configuration as follows:

- *Learning rate*: $\eta = 0.01$
- *Tree depth*: $d = 5$
- *Number of iterations*: $M = 30$

This configuration was selected based on multiple criteria: (1) minimal validation RMSE, (2) stable convergence behavior without oscillations, and (3) minimal overfitting as evidenced by small gaps between training and validation metrics. The moderate learning rate of 0.01 provides a good balance between convergence speed and optimization precision, avoiding both underfitting with larger rates and excessive computational cost with smaller rates. The tree depth of 5 captures sufficient feature interactions while remaining interpretable and avoiding excessive model complexity. The iteration count of 30 represents a point of diminishing returns where additional boosting rounds provided negligible improvements in performance.

### 2.4 Informative region selection

**2.4.1 Procedure overview.** To overcome sparsity in low-mutation cancers, we devised a two-stage selection: (1) iterative sampling and CatBoost training (1,000 repeats per subset size of 25, 50, 100, 150, and 200 windows), recording binary prediction outcomes; (2) Fisher's exact test with Benjamini–Hochberg FDR correction (q < 0.05) was used to identify regions whose repeated inclusion yielded significantly high accuracy.

**2.4.2 Optimal region number determination.** Using the significant windows, we retrained CatBoost models on subsets of increasing size (50, 100, 200, 300, 400, 500, 600 regions) to chart the relationship between the number of regions and TOO/COO prediction accuracy, thereby selecting the optimal subset scale.

## 3 Results

### 3.1 Benchmark evaluation demonstrates CatBoost superiority in multi-cancer TOO/COO classification

To demonstrate the superior performance of our CatBoost-based prediction model compared to existing methods, we evaluated tissue-of-origin (TOO) and cell-of-origin (COO) classification accuracy on a benchmark dataset of 137 samples spanning six cancer types. Fig 1 visualizes the prediction results obtained using Random Forest [5], XGBoost (COOBoostR) [18], and our CatBoost model [34]. The Nature study reported variance-explained values per cancer type and deemed a prediction correct if epigenetic marker signals fell within predefined thresholds. COOBoostR instead

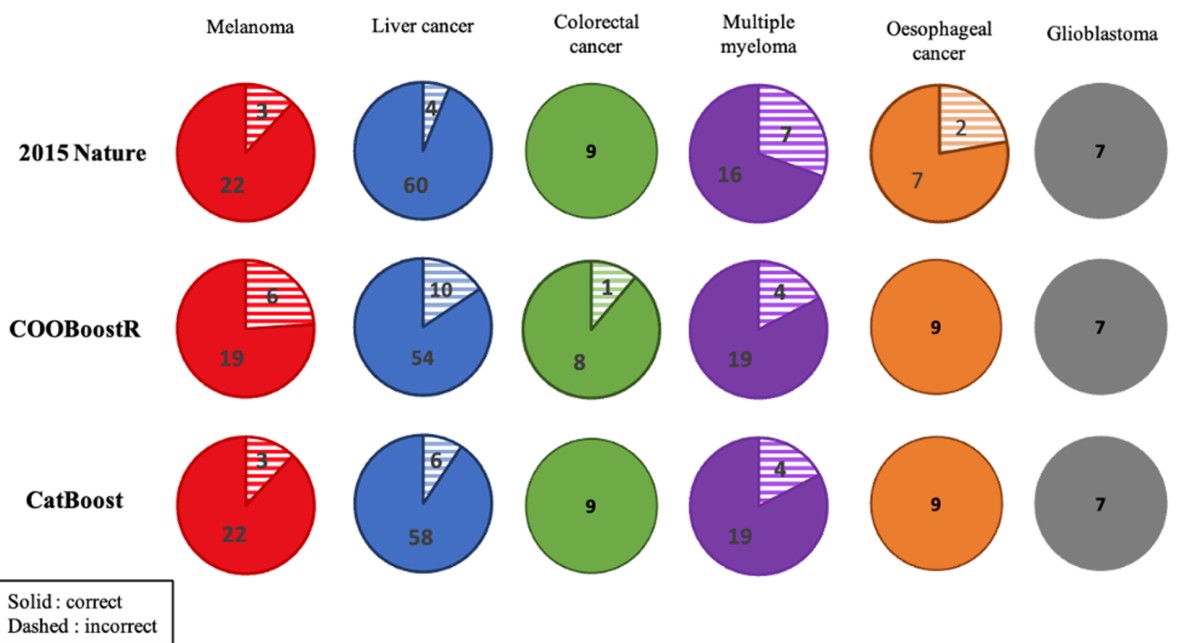

**Fig 1. Comparison of top-1 prediction accuracy for tissue-of-origin classification across models.** CatBoost (our study) outperformed or matched previous methods, including Random Forest (Nature, 2015) and COOBoostR (XGBoost-based), across all cancer types, demonstrating enhanced robustness and accuracy in tissue-of-origin prediction.

counted the number of exact top-1 epigenetic marker matches [18]. Our evaluation applies the stricter top-1 accuracy criterion across all methods to ensure a fair comparison.

Our CatBoost approach achieved equal or superior accuracy in all six cancers. In melanoma, CatBoost correctly classified 25/25 samples versus 22/25 for Random Forest and 19/25 for COOBoostR. Liver cancer accuracy reached ~90%, exceeding 84% for COOBoostR and outperforming Random Forest. Multiple myeloma classification was correct in 19/23 samples, matching or improving upon existing methods. For colorectal cancer, esophageal adenocarcinoma, and glioblastoma, CatBoost achieved 100% accuracy. These results establish CatBoost as a highly reliable alternative for TOO/COO prediction.

### 3.2 Informative regional subset analysis

We next assessed the effectiveness of Informative Region Selection by comparing CatBoost performance on regional subsets of size 50, 100, 200, 300, 400, 500, and 600 windows against whole-genome input (2,128 windows) (Table 1, Fig 2). Heatmap visualizations of selected regions per cancer are shown in Fig 3, (S1 Fig–S6 Fig).

High-mutation cancers (ESO, GBM) yielded 100% accuracy across all subset sizes, indicating robust signals that survive aggressive downsampling. Low-mutation cancers benefited most: melanoma accuracy rose from 88.0% (whole genome) to 92.0% (300 and 600 regions), and multiple myeloma from 82.6% to 87.0%. Colorectal cancer remained at 100% for several subset sizes but dipped to 88.9% when key regions were omitted. Liver cancer accuracy fell from 90.6% (whole genome) to 21.9% (50–100 regions) before recovering to 56.3–85.9% at larger subset sizes.

These findings demonstrate that Informative Region Selection significantly improves or maintains classification accuracy compared to random subsets, particularly in cancers with sparse mutation profiles, while reducing computational burden relative to whole-genome analysis.

**Table 1**. TOO/COO prediction accuracy by region selection strategy and subset size.

| Selection | Regions | CRC | ESO | GBM | ME | MM | RK |
|---|---|---|---|---|---|---|---|
| Random | Whole (2128) | 1.00 | 1.00 | 1.00 | 0.88 | 0.83 | 0.91 |
| | 50 | 0.13 | 0.15 | 0.24 | 0.24 | 0.08 | 0.09 |
| | 100 | 0.21 | 0.27 | 0.36 | 0.45 | 0.12 | 0.18 |
| | 200 | 0.39 | 0.59 | 0.53 | 0.56 | 0.16 | 0.32 |
| Informative | 50 | **1.00** | **1.00** | **1.00** | 0.80 | 0.44 | 0.22 |
| | 100 | 1.00 | 1.00 | 1.00 | 0.88 | 0.65 | 0.22 |
| | 200 | 0.89 | 1.00 | 1.00 | 0.88 | 0.83 | 0.56 |
| | 300 | 0.89 | 1.00 | 1.00 | **0.92** | 0.78 | 0.63 |
| | 400 | **1.00** | 1.00 | 1.00 | 0.88 | 0.78 | 0.75 |
| | 500 | 0.89 | 1.00 | 1.00 | 0.88 | 0.83 | 0.75 |
| | 600 | 0.89 | 1.00 | 1.00 | **0.92** | **0.87** | **0.86** |

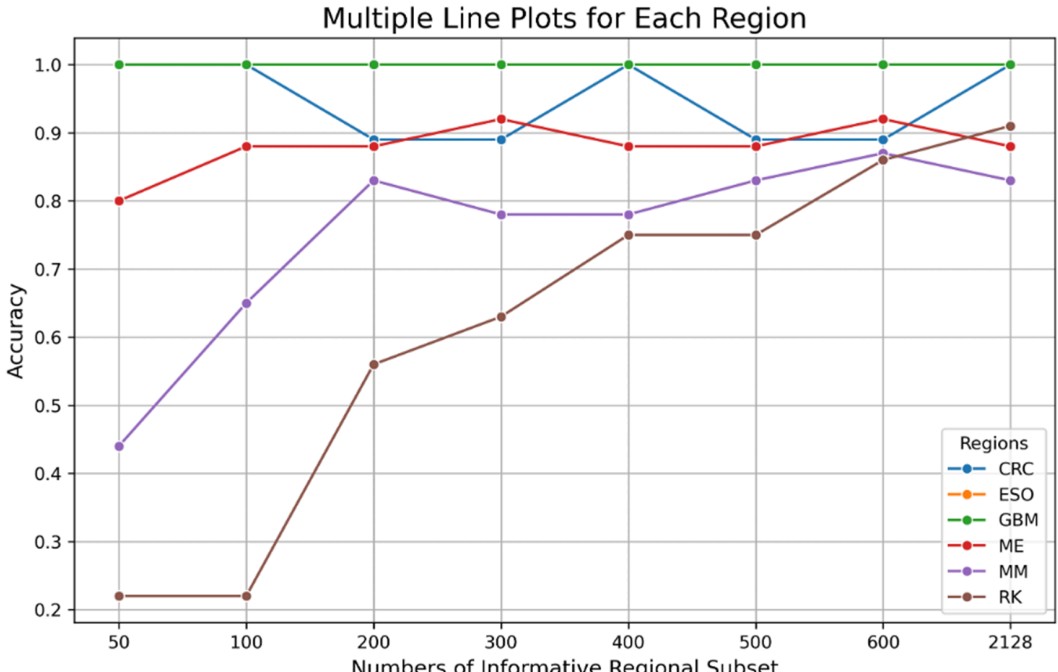

**Fig 2**. **Prediction accuracy by number of informative regions across six cancer types.** Most cancer types achieved over 60% accuracy with only 100 informative regions, demonstrating the efficiency of the regional subset strategy. Accuracy steadily improved with increasing region numbers. In liver cancer (RK), although initial accuracy was low, a consistent upward trend was observed, indicating effective learning with larger region sets.

### 3.3 Generalizability assessment in independent cohort (PCAWG)

To evaluate cross-cohort generalizability, we applied the selected regional subsets to 934 PCAWG samples across 14 lineages (Table 2, Fig 4). Gastrointestinal cancers (ESAD, COAD, STAD, READ) exhibited dramatic accuracy gains: starting at 28–46% with 50 regions, rising above 90% for 200+ regions, and reaching 100% for READ at 300 regions. Liver lineages (LIHC, LIRI, LICA, LINC) showed heterogeneous patterns, with cholangiocarcinoma (LICA) at 100% from 200 regions onward, whereas LIHC improved more gradually (up to 87.0% at 600 regions). Skin lineages improved from ~40% to >90% with 600 regions, and GBM accuracy climbed from 51.2% to 95.1%. Hematologic malignancies (MALY, CLLE, LAML) displayed variable performance, reflecting the complexity of predicting closely related cell origins.

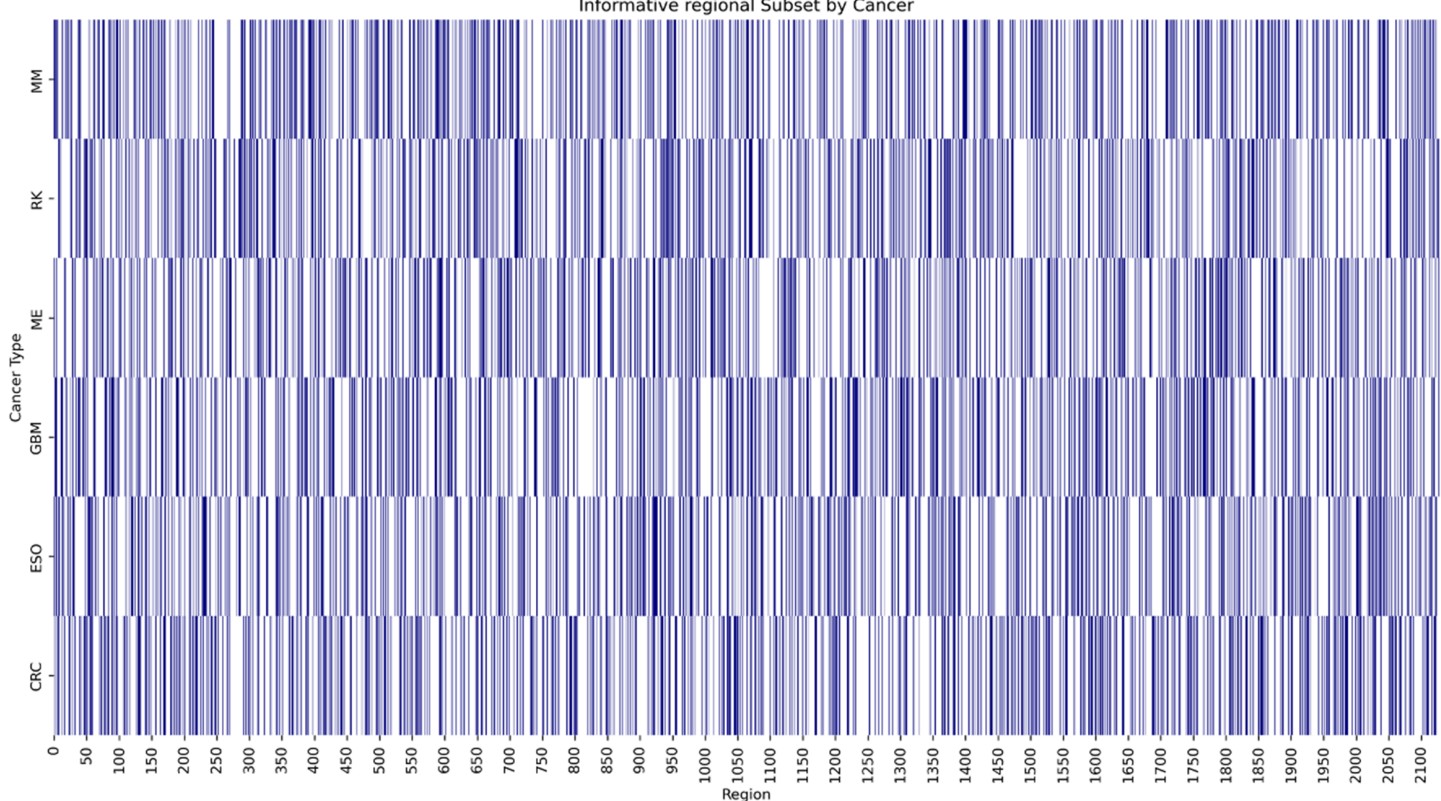

**Fig 3**. **Selection of 600 informative regional subsets out of 2,128 whole regions for each cancer type in the benchmark.** The figure shows the most frequently occurring genomic regions by aggregating the locations of regional subsets that successfully predicted the tissue or cell of origin for each cancer type.

**Table 2**. **PCAWG dataset TOO/COO prediction accuracy by informative regional subset size.**

| Regions | ESAD | STAD | COAD | READ | LIHC | LIRI | LICA | LINC | MELA | SKCM | GBM | MALY | CLLE | LAML |
|---------|------|------|------|------|------|------|------|------|------|------|-----|------|------|------|
| 50 | 0.45 | 0.28 | 0.46 | 0.25 | 0.17 | 0.19 | 0.50 | 0.26 | 0.40 | 0.42 | 0.51 | 0.23 | 0.29 | 0.12 |
| 100 | 0.76 | 0.39 | 0.74 | 0.63 | 0.17 | 0.23 | 0.33 | 0.23 | 0.70 | 0.71 | 0.83 | 0.54 | 0.41 | 0.28 |
| 200 | 0.94 | 0.67 | 0.87 | 0.88 | 0.52 | 0.60 | 1.00 | 0.65 | 0.81 | 0.79 | 0.85 | 0.63 | 0.58 | 0.40 |
| 300 | 0.92 | 0.72 | 0.85 | 0.94 | 0.63 | 0.67 | 1.00 | 0.77 | 0.76 | 0.79 | 0.90 | 0.68 | 0.61 | 0.28 |
| 400 | 0.92 | 0.82 | 0.96 | 0.88 | 0.80 | 0.74 | 1.00 | 0.74 | 0.81 | 0.87 | 0.93 | 0.74 | 0.73 | 0.37 |
| 500 | 0.95 | 0.80 | 0.94 | 1.00 | 0.80 | 0.81 | 1.00 | 0.77 | 0.83 | 0.92 | 0.93 | 0.80 | 0.70 | 0.44 |
| 600 | 0.96 | 0.82 | 0.94 | 1.00 | 0.87 | 0.80 | 1.00 | 0.74 | 0.87 | 0.92 | 0.95 | 0.86 | 0.73 | 0.28 |

Overall, Informative Region Selection consistently preserved or enhanced TOO/COO prediction accuracy across diverse independent cohorts, demonstrating its utility for efficient, interpretable cancer origin classification.

## 4 Discussion

In this study, we developed a regional subset–based framework to predict tissue-of-origin (TOO) and cell-of-origin (COO) across diverse cancer types, moving beyond conventional whole-genome approaches. A striking finding emerged from our analysis: two specific genomic intervals—the 921 region (chr2:155,326,172-156,326,171) and the 379 region (chr12:107,856,695-108,856,694)—were consistently and unanimously selected across all six cancer types in the

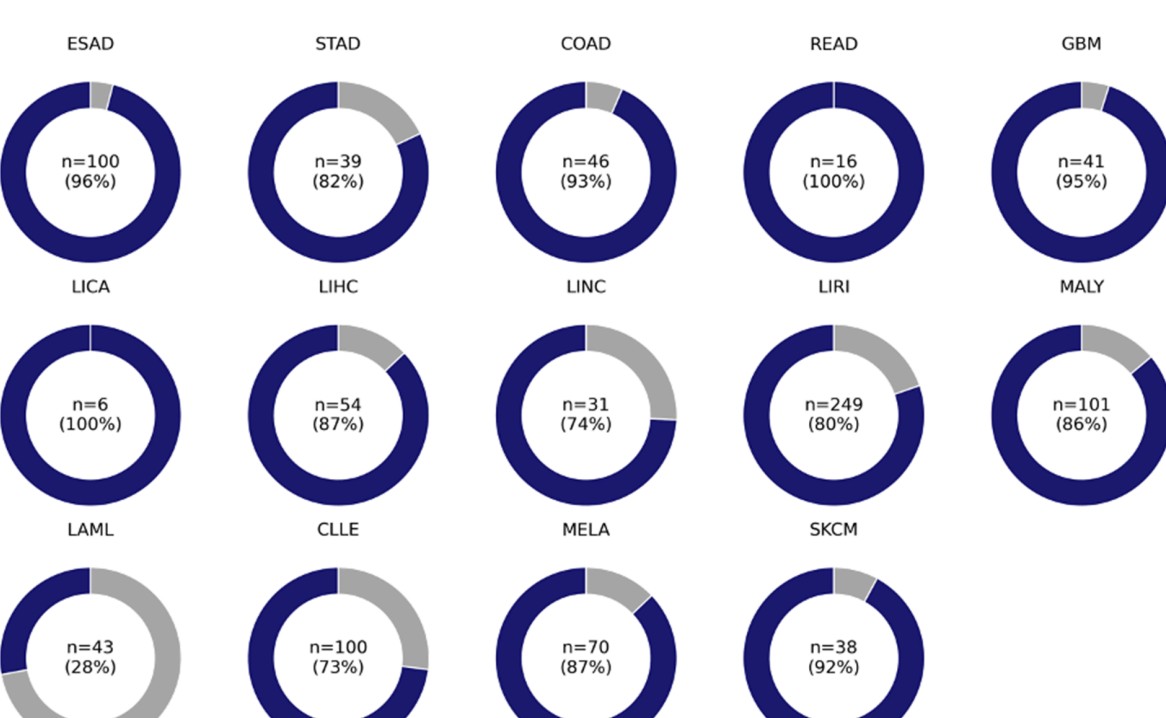

**Fig 4. PCAWG dataset accuracy donut chart.** The accuracy of Tissue/Cell of origin prediction using informative regional subsets is shown for various cancer types collected from similar tissues in the PCAWG dataset. Most similar cancer types demonstrated strong performance, but cancers like LAML, which arose from similar tissues but have different oncogenic mechanisms, were found to have poor utilization of the informative regional subsets.

benchmark cohort. This convergent selection across multiple cancer types suggests that these loci are not statistical artifacts but rather encode genuine tissue-associated biological signals. To understand why our CatBoost model converged on these regions, we conducted a detailed mechanistic analysis to validate that our predictive framework is grounded in real biological features rather than spurious correlations.

The predictive signal of the 921 region (chr2) appears to be rooted in mechanisms related to cell survival, transcriptional control, and stress response. Although a notable protein-coding gene, NR4A2 (Nuclear Receptor 4A2), is located immediately adjacent to the defined 1Mb boundary, its established function is highly pertinent to cancer biology. NR4A2 is a nuclear transcription factor that, when aberrantly regulated, is known to promote cancer progression by enhancing autophagy and inducing chemotherapy resistance. The consistent selection of this locus across multiple cancer types, therefore, suggests that intrinsic differences in survival and stress-evasion capabilities among various TOO phenotypes constitute a decisive genomic signature for classification. Furthermore, the presence of non-coding elements in these intervals indicates that predictive power may not rely exclusively on protein-coding genes; instead, lncRNAs and pseudogene/snRNA loci could act as tissue-specific transcriptional or chromatin-state markers that contribute indirect signals for origin classification. LINC01876 and nearby snRNA/pseudogene loci (e.g., RNU6-546P) are plausible markers of lineage-specific transcriptional programs; their roles are best interpreted as indicators of local transcriptional or epigenomic context that distinguish cellular origins.

In contrast, the 379 region (chr12) presents a dense cluster of protein-coding genes governing fundamental cellular processes, suggesting that the TOO/COO signature reflects inherent functional properties inherited from the cell of

origin. A significant functional cluster within this region relates to cellular motility and invasion. Genes such as CORO1C and SSH1 are essential components of the actin cytoskeleton remodeling pathway. CORO1C has been implicated in cell migration and invasion in several studies; however, its role as a driver in the tumor types examined here remains unestablished and requires further validation. Similarly, SSH1—which facilitates actin dynamics by regulating the actin-depolymerizing factor cofilin—may contribute to motility-related programs but also needs confirmatory evidence. The consistent selection of these motility-regulating genes across multiple cancer types strongly suggests that a tumor's innate tendency toward metastasis or migration is a major genomic feature distinguishing TOO and COO phenotypes. Additionally, the inclusion of ISCU (Iron-Sulfur Cluster Assembly Enzyme) may reflect the predictive signature of cancer metabolic reprogramming; ISCU-related mitochondrial dysfunction may contribute to tissue-specific metabolic phenotypes that the model captures. Finally, the roles of SART3 (RNA splicing and tumor antigen) and FICD (ER protein quality control) suggest that subtle differences in cellular housekeeping mechanisms—such as RNA processing fidelity and cellular stress management—are also predictive features captured by this region.

In summary, the robust selection of the 921 and 379 regions is strongly supported by a clear mechanistic analysis demonstrating that the genes within these loci are deeply integrated into core cancer pathways involving survival, motility, and metabolism. This finding is particularly significant because it demonstrates that our model is capturing genuine biological signals that distinguish cancer origins, rather than exploiting statistical noise or trivial patterns. Having established this biological foundation, we turn to examining how our CatBoost model leverages these mechanisms to achieve high predictive accuracy, while also identifying contexts where performance varies.

Our CatBoost model achieved high overall accuracy in predicting tissue and cell of origin across the six cancer types in the benchmark cohort. However, we observed notable variability in performance among liver cancer subtypes, providing insights into how the mechanistic features identified above translate into practical model behavior. Hepatocellular carcinoma (LIHC) and intrahepatic cholangiocarcinoma (LICA) were predicted with high accuracy, whereas the LIRI and LINC subcohorts showed diminished performance. To understand this discrepancy and connect it to the underlying biological mechanisms, we compared tumor mutational burden (TMB) distributions among these subtypes. LICA samples exhibited relatively uniform TMB (mean = 6.68; range 3.462–13.95), whereas LINC samples had lower and more homogeneous TMB (mean = 4.38; range 1.449–8.346). In contrast, LIRI displayed broad TMB variation (mean = 6.14; range 0.156–75.604) (S1 Table). These differences indicate that TMB magnitude and consistency critically impact model learning: the high heterogeneity observed in LIRI may obscure the consistent mutational patterns and metabolic signatures captured within the 921 and 379 regions, making it difficult for the model to identify reliable cell-of-origin markers. Conversely, the uniform and elevated TMB in LICA facilitates accurate classification because the biological signals encoded in these regions remain consistent and distinguishable across the tumor population. This observation underscores that the mechanistic features we identified (survival pathways, motility-related genes, and metabolic regulators) are most effective as classification signals when tumor populations maintain reasonable internal consistency.

Our region-selection approach generalized effectively to an independent PCAWG cohort, maintaining high accuracy in cancer types that were represented during model training. This cross-cohort validation strengthens our confidence that the 921 and 379 regions capture generalizable biological features rather than artifacts specific to our benchmark dataset. However, predictive performance declined for lineages not included in model training—such as acute myeloid leukemia (LAML)—highlighting an important limitation. This performance drop for untrained cancer types suggests that while the biological mechanisms we identified (cell survival, motility, and metabolism) are indeed central to cancer identity, the specific manifestations of these mechanisms may vary significantly across diverse lineages. Consequently, extending our framework to rare or highly heterogeneous malignancies will require lineage-specific or more granular subset selection strategies that capture the unique genomic signatures of each cancer type.

These results demonstrate that biologically informed regional subset selection can reduce input dimensionality, mitigate data sparsity, and capture key epigenetic–mutation associations, thereby enhancing TOO/COO prediction accuracy. This work moves beyond presenting a mere predictive model, as it successfully identifies the essential genomic regions

involved in establishing cancer identity, with clear mechanistic explanations for why these regions matter. The consistently selected loci represent highly promising targets for focused functional genomic studies that can validate the roles of identified genes and pathways in determining cancer origin. Future work will focus on refining selection procedures to account for intra-tumor heterogeneity, conducting experimental validation of the biological mechanisms identified in this analysis, and developing lineage-specific frameworks for cancer types not represented in the current training set. By integrating computational predictions with functional studies and expanding our approach to encompass broader cancer lineages, we can develop more mechanistically informed and clinically actionable frameworks for cancer origin determination, ultimately supporting their diagnostic and therapeutic applications.

## Supporting information

**S1 Fig. Occurrence frequency of each genomic region in random regional subsets yielding correct predictions (Liver cancer).**
(PNG)

**S2 Fig. Occurrence frequency of each genomic region in random regional subsets yielding correct predictions (Melanoma).**
(PNG)

**S3 Fig. Occurrence frequency of each genomic region in random regional subsets yielding correct predictions (Multiple Myloma).**
(PNG)

**S4 Fig. Occurrence frequency of each genomic region in random regional subsets yielding correct predictions (Glioblastoma).**
(PNG)

**S5 Fig. Occurrence frequency of each genomic region in random regional subsets yielding correct predictions (Esophagus cancer).**
(PNG)

**S6 Fig. Occurrence frequency of each genomic region in random regional subsets yielding correct predictions (Colorectal cancer).**
(PNG)

**S1 Table. Summary of cancer types sourced from PCAWG.**
(PNG)

## Author contributions

**Conceptualization:** Sungmin Yang.

**Data curation:** Sungmin Yang.

**Formal analysis:** Sungmin Yang.

**Investigation:** Sungmin Yang.

**Methodology:** Sungmin Yang.

**Project administration:** Sungmin Yang.

**Supervision:** Hong-Gee Kim.

**Validation:** Sungmin Yang.

**Visualization:** Sungmin Yang.

**Writing – original draft:** Sungmin Yang.

**Writing – review & editing:** Sungmin Yang.

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
