## [Decision Letter · Decision Letter 0]

28 Sep 2025

PONE-D-25-42292Biologically-Informed Regional Subset Analysis with CatBoost for Robust Tissue-of-Origin PredictionPLOS ONE

Dear Dr. Kim, Thank you for submitting your manuscript to PLOS ONE. After careful consideration, we feel that it has merit but does not fully meet PLOS ONE’s publication criteria as it currently stands. Therefore, we invite you to submit a revised version of the manuscript that addresses the points raised during the review process.

We look forward to receiving your revised manuscript.

Kind regards,

Yogendra Kumar Prajapati, Ph.D.

Academic Editor

PLOS ONE

Journal Requirements:

This work was supported by the National Research Foundation of Korea (NRF) grant funded by the Korea government (Ministry of Science and ICT) (Grant No. RS-2023-00268071).

Additional Editor Comments :

Based on the reviewers' comments, the manuscript is under minor revision. The authors are advised to revise the manuscript accordingly.

Reviewers' comments:

Reviewer's Responses to Questions

**Comments to the Author**

1. Is the manuscript technically sound, and do the data support the conclusions?

Reviewer #1: Yes

Reviewer #2: Yes

2. Has the statistical analysis been performed appropriately and rigorously?

Reviewer #1: No

Reviewer #2: N/A

3. Have the authors made all data underlying the findings in their manuscript fully available?

Reviewer #1: No

Reviewer #2: Yes

4. Is the manuscript presented in an intelligible fashion and written in standard English?

Reviewer #1: No

Reviewer #2: Yes

5. Review Comments to the Author

Reviewer #1: In this paper reports Biologically-Informed Regional Subset Analysis with CatBoost for Robust Tissue-of Origin Prediction. The concept in general is interesting and the results presented by the authors are interesting and sound. However, before accepting this work for publication, I would like to suggest the authors further revise the paper after taking into account the following comments.

1. Why authors used the only CatBoost machine learning technique only.

2. Much more discussion about the results should be given in this paper, especially the author needs to provide enough physicals mechanism analysis about the results.

3. The authors should clarify the mathematical framework used to the CatBoost model.

4. The figures in the manuscript are blurred and need improvement.

5. In the manuscript, there are grammatical and spelling mistakes. The language is comprehensive and coherent, but there are some mistakes.

Manuscript can be accepted if the authors will incorporate the my comments.

Reviewer #2: Manuscript Number: PONE-D-25-42292

Reviewer’s Comments: The manuscript titled “Biologically-Informed Regional Subset Analysis with CatBoost for Robust Tissue-of-Origin Prediction” presents a CatBoost prediction model for precision oncology and the diagnosis of cancers. This approach achieved a 4% gain in melanoma accuracy and a 4.4% gain in multiple myeloma and perfect (100%) accuracy in high‑mutation cancers such as esophageal adenocarcinoma and glioblastoma with as few as 50 informative regionals subset. On considering all things, these findings show that biologically guided regional subset 168 selection can improve TOO/COO prediction accuracy by lowering input dimensionality, mitigating data sparsity, and capturing important 169 epigenetic–mutation correlations. I am convinced with the method applied for the analysis. However, before publishing, it requires following minor revision to incorporate the all given suggestions.

1. To enhance the comprehensiveness of your paper, it would be beneficial to include at least five previous studies for comparative analysis, thereby strengthening its position relative to existing literature on the same topic.

2. The novelty of the research work is not clear. The authors are suggested to provide suitable novelty aspects.

3. Can you provide justification for claiming that your work over existing work in literature?

4. Labels of Fig. 3 are not clearly visible. Authors need to revise it.

5. Most of the cited reference work is before 2020. It is suggested to add latest relevant references.

6. PLOS authors have the option to publish the peer review history of their article (what does this mean?). If published, this will include your full peer review and any attached files.

Reviewer #1: No

Reviewer #2: **Yes: **Dr. Sarika Pal

---

## [Author Response · Author response to Decision Letter 1]

17 Oct 2025

Manuscript ID: PONE-D-25-42292

Title: Biologically-Informed Regional Subset Analysis with CatBoost for Robust Tissue-of-Origin Prediction

Journal: PLOS ONE

Type: Response to Reviewers

Date: 15 Oct, 2025

Dear Editor and Reviewers,

We sincerely thank the Academic Editor and the reviewers for their constructive comments and valuable suggestions that helped us improve the quality and clarity of our manuscript.

We have carefully revised the manuscript in accordance with all comments and suggestions. Below, we provide our detailed responses to each point.

All changes have been highlighted in the revised manuscript using track changes.

Response to Reviewer 1

Comment 1:

Why authors used the only CatBoost machine learning technique only.

Response:

We thank the reviewer for this important question. In previous studies utilizing tumor mutation and methylation data, algorithms such as Random Forest and XGBoost were commonly applied. These models share the characteristic of being tree-based ensemble methods, primarily aimed at extracting features that can explain model predictions. Moreover, in recent biomedical data analyses, CatBoost has emerged as a widely used interpretable model. Accordingly, we compared the performance of CatBoost with Random Forest and XGBoost in our dataset and observed superior performance. Therefore, we applied our proposed informative regional subset analysis using CatBoost. This explanation has been added to the Introduction section (pages 1, lines 30–36).

Comment 2:

Much more discussion about the results should be given in this paper, especially the author needs to provide enough physical mechanism analysis about the results.

Response:

We appreciate the reviewer’s insightful comment. In response, we have substantially expanded the Discussion section to include not only a description of the common regions identified as important in our analysis across cancer types, but also detailed analyses of their chromosomal locations. Furthermore, we identified the transcripts within these regions using the UCSC Genome Browser and incorporated relevant biological information and our interpretations. These additions provide a more comprehensive discussion of the potential biological mechanisms underlying our findings (Discussion section, pages 11–12, lines 243–297).

Comment 3:

The authors should clarify the mathematical framework used to the CatBoost model.

Response:

Following the reviewer’s suggestion, we have added a detailed explanation of the mathematical framework underlying the CatBoost model. This includes the fundamental principles of gradient boosting and the base model formulation. This new subsection has been incorporated into the Methods section of the revised manuscript. (pages 3–4, lines 120–144)

Comment 4:

The figures in the manuscript are blurred and need improvement.

Response:

We have replaced all blurred figures with high-resolution versions and ensured that the axis labels, legends, and annotations are clearly visible in the revised manuscript. In particular, Figure 3, which appeared noticeably less clear than the others, has been replaced with a high-resolution version to improve visual quality. (Result section, page 8, Figure 3)

Comment 5:

In the manuscript, there are grammatical and spelling mistakes. The language is comprehensive and coherent, but there are some mistakes.

Response:

We have carefully proofread the entire manuscript and corrected all grammatical and typographical errors. The revised text has also been checked by a professional English editing service.

Response to Reviewer 2

Comment 1:

To enhance the comprehensiveness of your paper, it would be beneficial to include at least five previous studies for comparative analysis, thereby strengthening its position relative to existing literature on the same topic.

Response:

We thank the reviewer for this important suggestion. To enhance the comprehensiveness of our manuscript, we have now compared the datasets, analysis methods, and results of recent related studies. We also summarized the limitations of prior work to highlight the novelty and necessity of our study. These revisions have been added to the Introduction section (pages 1–2, lines 30–63).

Comment 2:

The novelty of the research work is not clear. The authors are suggested to provide suitable novelty aspects.

Response:

We thank the reviewer for this constructive comment. To clarify the novelty of our study, we have explicitly highlighted the main contributions of our work, including a concise description of the novel aspects of our approach and an expanded literature review to contextualize our findings relative to existing studies. These revisions are included in the Introduction section (pages 2, lines 64–88). (pages 2, lines 64–88)

Comment 3:

Can you provide justification for claiming that your work over existing work in literature?

Response:

We appreciate the reviewer’s suggestion. To emphasize the advantages of our study over existing work, we have added a discussion highlighting the limitations of previous studies and the biological and computational strengths of our approach in the Introduction section.(pages 2, lines 48–63)

Comment 4:

Labels of Fig. 3 are not clearly visible. Authors need to revise it.

Response:

We have revised Figure 3 to improve label clarity, font size, and contrast for better readability. (Result section, page 8, Figure 3)

Comment 5:

Most of the cited reference work is before 2020. It is suggested to add latest relevant references.

Response:

We thank the reviewer for this valuable suggestion. In response, we have added more than 10 recent relevant studies to the manuscript. These additions help to highlight the novelty of our work, differentiate it from existing studies, and clarify the limitations of prior research, thereby emphasizing the advantages of our approach. These revisions are included in the Introduction section. (pages 1–2, lines 30–63)

Conclusion

We thank the reviewers and editor once again for their time and constructive feedback.

We believe that the revised version has been significantly improved and hope that it now meets PLOS ONE’s publication standards.

Sincerely,

[Sungmin Yang / Professor Hong-Gee Kim]

---

## [Decision Letter · Decision Letter 1]

4 Nov 2025

Biologically-Informed Regional Subset Analysis with CatBoost for Robust Tissue-of-Origin Prediction

PONE-D-25-42292R1

Dear Dr.Hong-Gee Kim,

We’re pleased to inform you that your manuscript has been judged scientifically suitable for publication and will be formally accepted for publication once it meets all outstanding technical requirements.

Kind regards,

Yogendra Kumar Prajapati, Ph.D.

Academic Editor

PLOS ONE

Additional Editor Comments (optional):

Based on the reviewer's decision, the manuscript may be accepted for publication.

Reviewers' comments:

Reviewer's Responses to Questions

**Comments to the Author**

1. If the authors have adequately addressed your comments raised in a previous round of review and you feel that this manuscript is now acceptable for publication, you may indicate that here to bypass the “Comments to the Author” section, enter your conflict of interest statement in the “Confidential to Editor” section, and submit your "Accept" recommendation.

Reviewer #1: All comments have been addressed

Reviewer #2: All comments have been addressed

2. Is the manuscript technically sound, and do the data support the conclusions?

Reviewer #1: Yes

Reviewer #2: Partly

3. Has the statistical analysis been performed appropriately and rigorously?

Reviewer #1: I Don't Know

Reviewer #2: N/A

4. Have the authors made all data underlying the findings in their manuscript fully available?

Reviewer #1: No

Reviewer #2: Yes

5. Is the manuscript presented in an intelligible fashion and written in standard English?

Reviewer #1: Yes

Reviewer #2: Yes

6. Review Comments to the Author

Reviewer #1: Authors are successfully incorporated the my comments. Manuscript can be accepted for the publication.

Reviewer #2: (No Response)

7. PLOS authors have the option to publish the peer review history of their article (what does this mean?). If published, this will include your full peer review and any attached files.

Reviewer #1: No

Reviewer #2: No

---

## [Editor Report · Acceptance letter]

PONE-D-25-42292R1

PLOS ONE

Dear Dr. Kim,

I'm pleased to inform you that your manuscript has been deemed suitable for publication in PLOS ONE. Congratulations! Your manuscript is now being handed over to our production team.

Kind regards,

on behalf of

Dr. Yogendra Kumar Prajapati

Academic Editor

PLOS ONE